# Reviewing the Potential Links between Viral Infections and TDP-43 Proteinopathies

**DOI:** 10.3390/ijms24021581

**Published:** 2023-01-13

**Authors:** Zerina Rahic, Emanuele Buratti, Sara Cappelli

**Affiliations:** 1Saadiyat Island Campus, New York University Abu Dhabi, Abu Dhabi P.O. Box 129188, Saadiyat Island, United Arab Emirates; 2International Centre for Genetic Engineering and Biotechnology (ICGEB), AREA Science Park, Padriciano 99, 34149 Trieste, Italy

**Keywords:** TDP-43, RNA-binding proteins, RNA metabolism, viral entry, viral replication, viral latency, neuronal dysfunctions

## Abstract

Transactive response DNA binding protein 43 kDa (TDP-43) was discovered in 2001 as a cellular factor capable to inhibit HIV-1 gene expression. Successively, it was brought to new life as the most prevalent RNA-binding protein involved in several neurological disorders, such as amyotrophic lateral sclerosis (ALS) and frontotemporal lobar degeneration (FTLD). Despite the fact that these two research areas could be considered very distant from each other, in recent years an increasing number of publications pointed out the existence of a potentially important connection. Indeed, the ability of TDP-43 to act as an important regulator of all aspects of RNA metabolism makes this protein also a critical factor during expression of viral RNAs. Here, we summarize all recent observations regarding the involvement of TDP-43 in viral entry, replication and latency in several viruses that include enteroviruses (EVs), Theiler’s murine encephalomyelitis virus (TMEV), human immunodeficiency virus (HIV), human endogenous retroviruses (HERVs), hepatitis B virus (HBV), severe acute respiratory syndrome coronavirus 2 (SARS-CoV-2), West Nile virus (WNV), and herpes simplex virus-2 (HSV). In particular, in this work, we aimed to highlight the presence of similarities with the most commonly studied TDP-43 related neuronal dysfunctions.

## 1. Introduction

Transactive response DNA binding protein 43 kDa (TDP-43), encoded by the *TARDBP* gene, belongs to the family of heterogeneous nuclear ribonucleoproteins (hnRNPs) that serve multiple roles in the generation and processing of RNA. Its critical role in RNA processing occurs through its interaction with many ribonucleoprotein complexes (spliceosome, Drosha, poly-adenylation, stress granule, and translational complexes) in the nucleus and cytoplasm. Through all these interactions, TDP-43 is now known to regulate many cellular processes that include all aspects of gene expression, from transcription to RNA maturation (both coding and non-coding), transport, stability, and eventual translation [1,2,3,4,5,6,7].

This ubiquitously expressed RNA-binding protein predominantly resides in the nucleus although a minor proportion is always shuttling back and forth to the cytoplasm together with its bound RNAs. However, cellular stresses are known to transiently increase the amount of TDP-43 into the cytoplasm and assemble in stress granules [8]. As a result, many cellular alterations that lead to disruption of nucleocytoplasmic trafficking, prolonged stress, and changes in liquid–liquid phase separation can lead to TDP-43 accumulation in insoluble aggregates that are located mostly in the cytoplasm of glia and degenerating neurons in the central nervous system (CNS) [9]. The TDP-43 in these aggregates is variably modified and subjected to cleavage, ubiquitination, and several other post-translational modifications that include phosphorylation, acetylation, and sumoylation [10,11,12]. All these modifications represent pathological hallmarks of amyotrophic lateral sclerosis (ALS), frontotemporal lobar degeneration (FTLD) as well as other types of conditions such as traumatic brain injury [10,13]. In general, however, aberrant cleavage and cytoplasmic aggregation of TDP-43 are identified as molecular signatures for most forms of ALS and FTLD and have been shown to contribute significantly to disease progression [10]. Following aggregation in insoluble forms, the loss of “active” TDP-43 in the cellular pool is thought to lead to abnormalities of splicing and RNA metabolism with subsequent neuronal dysfunction [14]. Very recently, this has been clearly observed in patients by the detection of “cryptic” exons inclusion in important genes, such as UNC13A and STMN2, where the absence of TDP-43 during the splicing process induces the spliceosome to include pre-mRNA sequences in the mature mRNA which in normal conditions would never have been included (with consequential loss of function effects) [15,16].

The cellular toxicity that results from TDP-43 aggregation pathology is thought to initiate a cascade of downstream toxic effects that not only occur within the initiating cell but also can be spread between vulnerable cell types and to other regions of the nervous system [17]. This transmission through the tissue may involve prion-like properties of TDP-43, in which pathologically misfolded protein templates the propagation of aggregation pathology [17,18,19]. 

From a historical point of view, it is interesting to note that the first identification of TDP-43 occurred in the mid-nineties when it was hypothesized to bind the DNA transactivation–responsive (TAR) sequence of the human immunodeficiency virus type 1 (HIV-1) and inhibit viral transcription; hence, the name of TDP-43, where 43 refers to the molecular weight of the wild-type protein based on its primary amino acid sequence [1]. However, this transcriptional inhibitory activity was not confirmed in later studies [20], although more recent studies have reported the capability of TDP-43 to influence cell permittivity to HIV-1 infection [21] and latency [22].

In parallel and interestingly, enterovirus-induced pathology has also been widely studied in the context of ALS and TDP-43 [23]. However, despite great efforts in this direction, the clinical data are controversial, probably due to the inconsistency and differences in viral detection techniques, disease stage of sample collection, as well as the potential virus-triggered “prion-like mechanism”. Most importantly, however, although a causal relationship between chronic viral infection and ALS development remains to be established, many recent observations have reported that TDP-43 can play a role in many viral infections. Therefore, this review article aimed to describe the novel potential relationship between TDP-43 and enteroviruses (EVs), Theiler’s murine encephalomyelitis virus (TMEV), human immunodeficiency virus (HIV), human endogenous retroviruses (HERVs), hepatitis B virus (HBV), severe acute respiratory syndrome coronavirus 2 (SARS-CoV-2), West Nile virus (WNV), and herpes simplex virus-2 (HSV).

## 2. Viral Infections and TDP-43

### 2.1. Enteroviruses (EVs)

This is probably the most studied virus with regards to ALS because recent studies have suggested that the mechanisms underlying cellular dysfunctions in ALS appear to be closely related to enterovirus-induced pathology [24,25]. Therefore, there is a critical need for evidence to support the suspected contribution of enteroviruses (EVs), a family of positive stranded RNA viruses, to the development of ALS. 

Recently, a molecular link has been demonstrated between EV infection and ALS pathogenesis as the consequence of RNA-processing defects [26,27,28], impairment in nucleocytoplasmic transport [29,30], neuroinflammation [31,32], and disrupted protein quality control [33,34]. An explanation behind these altered cellular processes arises by the fact that the replication of EVs occurs in cytoplasm, where many RBPs, including TDP-43, are hijacked during EV infection [35]. For example, dysregulation of TDP-43 during Coxsackievirus B3 (CVB3) infection was found to be involved in viral pathogenesis and infectivity [26]. Six serotypes of group B coxsackievirus are currently known and CVB3 is responsible for several impairment mobilities with a wide spectrum of symptoms, including pancreatitis, myocarditis, aseptic meningitis, and even juvenile diabetes [36]. It has been demonstrated that CVB3 infection causes a cytoplasmic redistribution and protein aggregation of TDP-43, leading, in turn, to the loss-of-function of endogenous TDP-43. This process was found to be mediated by the viral protease 3C, whereby creating a N-terminal cleavage fragment of TDP-43 is able to interfere with the splicing properties of TDP-43 itself [26]. 

### 2.2. Theiler’s Murine Encephalomyelitis Virus (TMEV)

Theiler’s murine encephalomyelitis virus (TMEV) is a single-stranded RNA murine virus belonging to the family *Picornaviridae* and closely related to *Cardiovirus* subgroups [37]. This virus is highly virulent, and it is the causative agent of acute encephalitis in mice [38]. On the basis of its genetic characteristics and tropism, this virus has been largely used as a model for multiple sclerosis [39].

Recently, Masaki and collaborators have pointed out a potential role of TDP-43 in TMEV infection. In their study, it has been observed that cytoplasmic mislocalization and phosphorylation of TDP-43, along with cleavage into products similar in size to those found in ALS, occurred in TMEV-infected cultured cells as well as in neuronal and glial cells of TMEV-infected mice [40]. This TDP-43 mislocalization was presumably supported by the fact that TDP-43 can bind to mRNAs encoding myelin genes and its depletion can lead to demyelination and neuronal death [41], as well as by the concomitant action of further inflammatory stimuli such as the expression of tumor necrosis factor-α [42].

In addition, in virus-induced demyelinated regions, other RNA-binding proteins were found to be mislocalized in glial cells, including oligodendrocytes [40]. Taken together, these results have suggested that the mislocalization of RNA-binding proteins in TMEV infections could disrupt cellular splicing and mRNA translation, thereby contributing to the observed neuronal dysfunction and death [40].

### 2.3. Human Immunodeficiency Virus (HIV)

Human immunodeficiency virus (HIV) is a retrovirus belonging the family of *Retroviridae* and *Orthoretrovirinae* subfamily [43]. Two subtypes are described in literature, based on genetic and viral antigen differences: HIV type 1 (HIV-1) and HIV type 2 (HIV-2). Structurally, the HIV genome consists of two identical single-stranded RNA molecules that are enclosed within the core of the provirus particle.

Over the years, HIV-1 infection had a global impact on social and economy affairs, due to its association with several illnesses, including acquired immune deficiency syndrome (AIDS) [44], cardiovascular disease [45], bone dysfunction [46], hepatic, and renal impairment [47,48].

From a biological point of view, a multitude of host factors has been found to be involved in HIV pathology. Among these, TDP-43 was initially identified in 1995 as a transcriptional repressor of HIV-1 gene expression through the binding of the trans-activation response element (TAR) DNA sequence at the level of the HIV-1 long terminal repeat (LTR) promoter [1]. Unfortunately, this relationship was challenged in later studies which failed to demonstrate the capability of TDP-43 to modulate the HIV-replication in vivo [20]. In this study, it was shown that modulating TDP-43 expression in several cellular model of HIV infection does not represent a viable strategy to prevent transcription from viral genome. Indeed, the HIV-replication in T cell and macrophages was found to be independent on TDP-43 expression [20].

Nevertheless, some interesting connections between HIV-1 and TDP-43 have emerged in recent years. In fact, TDP-43 has been shown to be potentially involved in HIV-1 latency and cell permittivity. Specifically, Rathore and collaborators have demonstrated that reactivation of HIV-1 was promoted by removal of the steric hindrance posed by TDP-43 at the level of HIV-1 LTR promoter [22]. In addition, the Valenzuela–Fernández group have found that the silencing of TDP-43 was able to reduce the expression of the antiviral enzyme histone deacetylase 6 (HDAC6), both at the mRNA and protein level, increasing the fusogenic and infection activities of HIV-1 [21].

Overall, therefore, the relationship between HIV and TDP-43 still remains complex and concomitant pathologies may contribute to influence viral entry, disease progression and eventually neuronal susceptibility.

In this context, it is interesting to mention that a link between ALS-related TDP-43 pathology and HIV infection seems to be consistent with the re-activation of human endogenous retrovirus-K (HERV-K), indicating that the mechanisms behind TDP-43 dysfunction and neuronal inflammation are similar in both ALS and retroviral pathology [49,50].

### 2.4. Human Endogenous Retrovirus K (HERV-K)

Human endogenous retroviruses (HERV) are endogenous retroviral sequences that account for ca. 8% of human genome [51,52]. These viral elements are derived from an ancestral infection of germ-line cells by exogenous retrovirus and conserve structural similarities with exogenous retroviruses, such as HIV [51]. HERV re-activation has been progressively associated with neurological disfunction, e.g., schizophrenia [53], multiple sclerosis [54,55] and motor neuron disease [56], as well as cancer [55,57]. Notably, a novel link between TDP-43 and HERV-K has been provided by two articles published few years ago. Basically, it was reported that the disruption of the HERV-K env gene was associated with a significant decrease of TDP-43 expression, at both the mRNA and protein level [58] and that over-expression of ALS-associated TDP-43 mutants was reported to significantly increase HERV-K viral protein accumulation [50]. This scenario highlights the interdependence of HERV-K expression to TDP-43 and vice versa, suggesting that there is a tight regulation of both factors especially in neuronal cells.

### 2.5. Hepatitis B Virus (HBV)

Hepatitis B virus (HBV) is a hepatotropic and small DNA virus member of the *Hepadnaviridae* family, with a partially double-stranded relaxed circular DNA (rcDNA) genome which may cause severe liver diseases, such as liver cirrhosis and hepatocellular carcinoma [59].

In order to understand the molecular mechanisms behind the control of HBV replication and pathogenesis, a plethora of studies have been performed to analyze potentially implicated cellular factors, including hnRNPs [60,61]. In this regard, it has been recently suggested that TDP-43 may play a role in promoting HBV infection through several mechanisms involving DNA and RNA binding. In particular, Mokokha and collaborators have described TDP-43 as an important host factor capable to facilitate HBV gene expression by stimulating transcription from the HBV core promoter, by inhibiting the pre-genomic (pg) RNA splicing, and by participating to the assembly of protein complexes implicated in transcriptional and post-transcriptional stages of the virus life cycle [61]. Overall, this study has provided a valuable insight into the interaction between host cells and HBV and the authors have actually suggested to exploit TDP-43 as a potential target for novel anti-HBV therapies [61].

### 2.6. Severe Acute Respiratory Syndrome Coronavirus 2 (SARS-CoV-2)

Severe acute respiratory syndrome coronavirus 2 (SARS-CoV-2) is a positive-sense, single-stranded RNA virus that belongs to the *Coronaviridae* family and has been recently identified as the etiological agent of coronavirus disease 2019 (COVID-19) [62].

Most interestingly, in addition to respiratory disease, several abnormalities in the central nervous system such as cognitive decline, neuronal autoimmune disease, and delirium have been observed in patients suffering from SARS-CoV-2 pathology [63]. As a result, it has been suggested that coronaviruses contribute to trigger these neuropathologies through different mechanisms, including direct neurotoxic effects on synaptic transport [64], as well as neuronal inflammation [65].

In 2021, SARS-CoV-2 spike protein S1 was found to be linked to several amyloidogenic proteins, including tau and TDP-43 [66]. In this regard, TDP-43 RRMs (RRM1 and RRM2) were found to form eleven H-bonds and one salt bridge interaction with the viral S1 protein [66]. Moreover, hyperphosphorylation and subsequently aggregated inclusion of TDP-43 have been detected in the brain of SARS-CoV-2 infected patients [67]. Finally, while several neurodegeneration markers (YKL40, NCAM-1, CCL23) were elevated in COVID-19 survivors, the serum levels of TDP43 were found to diminish after admission to recover at 28 days in survivors [68]. In addition, ferritin, which is used to gauge the degree of inflammation in several inflammatory processes and COVID-19, was described to be the main marker of inflammation that correlates with TDP43 [68]. Overall, these results suggest that alteration of different neuronal factors, including TDP-43, may favor the establishment of neurodegenerative symptoms observed in individuals with acute COVID-19 disease.

### 2.7. West Nile Virus (WNV)

West Nile virus (WNV) is a *Flavivirus* with a positive-sense, single-stranded RNA genome that is responsible for severe neuroinvasive disease, such as meningitis and encephalitis [69,70]. In 2022, Constant and collaborators discovered several biomarkers involved in neurodegeneration in the serum of patients with WNV disease (WNVD) [71]. Among these factors, TDP-43 expression has been found to be significantly increased in WNVD patient serum [71]. Considering that in other similar viral infections accumulation of TDP-43 aggregates has been associated to a direct effect on viral replication [26], these researchers have hypothesized that the overexpression of TDP-43 in MNV infection could increase the chance to develop brain degenerative disorders [71].

### 2.8. Herpes Simplex Virus-2 (HSV-2)

Herpes simplex virus-2 (HSV-2) is an enveloped doubled-stranded DNA virus, subtype 2 of the herpes simplex viruses and together with HSV type 1 (HSV-1) it belongs to the *Alphaherpesvirinae* subfamily of the *Herpesviridae* family [72]. Despite HSV-1 and HSV-2 being mostly known as the causative agents of oral and genital ulcerative lesions, respectively, these viruses are also able to infect several tissues, including brain [72,73].

Specifically, HSV-2 has been associated with neurological disorders due to its capability of establishing long-lasting infection within nervous system [74]. In this context, therefore, it has been proposed in a recent study that HSV-2 latently infected neurons could express increased levels of endogenous TDP-43 in response to the presence of HSV latency-associated transcripts. This hypothesis was also corroborated by the identification of a multitude of potential TDP-43 binding sites along these sequences [75].

## 3. The Role of Neuroinflammation in Viral Infection and Neurodegeneration: A Potential Link with TDP-43 Dysfunction

Research interest in investigating the relationship between neuronal inflammation and neurodegeneration has been exponentially increased through the pass years. The establishment of neuronal dysfunctions frequently involve the release of numerous pro-inflammatory cytokines and the activation of apoptotic pathways leading to cell death [76]. Specifically, abnormal accumulation of pro-inflammatory cytokines, such as interleukins (IL)-1β [77], IL-6 [78], and transforming growth factor beta (TGF-β) [79], has been described surrounding the amyloid plaques of Alzheimer’s disease (AD) patients, as well as in serum and plasma of patients affected by neuropsychiatric disorder [80]. In addition, increasing levels of IL-4, IL-5, IL-10, IL-6, and tumor necrosis factor alpha (TNF-α) were detected in serum of ALS patients comparing to healthy subjects [81], and some of those factors were also linked to neuroinflammation in FTD [82,83,84]. Finally, it is important to note that an imbalance between pro- and anti-inflammatory molecules has been described during different viral infections that are occasionally associated with the Induction of neurological manifestations. For example, it has been widely demonstrated that the injection of pro-inflammatory cytokine TNF-α induces neuropathic pain in humans [85,86]. TNF-α is normally expressed at low levels in CNS, although microglia and astrocytes can release high amount of this factor during injury, neurodegenerative disorders, and infections [87]. For example, studies performed in mice have highlighted that injection of HIV envelope glycoprotein gp120 induced the activation of microglia and astrocytes with the consequent production of TNF-α at the spinal cord levels. This, in turn, triggered neuropathy symptoms similar to those seen in HIV-1 patients with neuropathic pain [88]. Another example of immunological neuropathy has been described in TMEV infection. TMEV is an RNA virus with tropism for nervous system where it establishes chronical demyelinating disorder in susceptible mouse strains similar to that observed in multiple sclerosis. During TMEV infection, it has been demonstrated that both microglia and macrophages stimulate the production and release of numerous interleukins, TNF-α, and Interferon gamma (IFN-γ) [89]. Neurological symptoms were also associated with the activation of endogenous retroviruses and more recently observed in patients affected by prolonged SARS-CoV-2 pathology. In these cases, the mechanisms behind those symptoms are linked to autoimmunity or chronic inflammation [90,91].

With this in mind, it is important to highlight that several immunoproteins are spatially and temporally modulated in the nervous system, as well as expressed in specific subsets of neuronal cells [92,93]. Moreover, the impairment of the cross-talk between neurons and glia has been described as a further consequence of neuroinflammatory cascades. In this respect, astrocytes (macroglia) play a fundamental role in maintaining neuronal milieu due to their neuroprotective and metabolic functions, such as controlling blood–brain barrier (BBB) integrity, modulating the extracellular amount of neurotransmitter and ions, the elimination of toxins from cerebrospinal fluid and providing trophic sources for neuron’s energy metabolism [94]. Notably, reactive astrocytes were found to be involved in the production of several neuronal mediators during viral infections. This process is also supported by microglia through the expression of IL-1, IL-6, TNF-α, and IFN-γ that in turn stimulates astrocyte proliferation throughout a defense mechanism namely astrogliosis [95].

In addition to these connections, defects in RBP functions due to mutations, post-translational modifications, and aggregation, may contribute to exacerbate and sustained this unhealthy process. In fact, it is well-known that RBPs are key players in the RNA metabolism especially in neurons where they are particularly expressed [96]. Recently, it has been demonstrated that pro-inflammatory mediators, such as IFN-γ and TNF-α, are able to trigger hnRNP A1 mislocalization and stress granule formation in murine primary neuron culture, with consequently neurite damage [97]. Specifically, SAFA (also known as hnRNP U) was found to be involve in protecting cells from viral infections, since it acts at a nuclear sensor for viral dsRNA and is able to trigger the activation of super-enhancer of anti-viral gene expression, such as IFNB1 [98]. Interestingly, the ability of hnRNPs to commonly regulate mRNA targets implicated in both brain functions and inflammatory pathways is recently emerged. siRNA mediated the depletion of hnRNP U, hnRNP D, hnRNP K, DAZAP1, hnRNP Q, hnRNP R, and TDP-43 in human neuroblastoma SH-SY5Y cell line pointed out the importance of RBP homeostasis in modulating both processes and providing further evidence of their connection [99,100]. Among the genes commonly regulated by TDP-43 and other hnRNPs is important to mention TNF-α (described above) and intercellular adhesion molecule 1 (ICAM-1, or CD54). Although *TNF-α* mRNA was found to be upregulated in SH-SY5Y treated with siRNA against TDP-43 [99], in monocytes TNF-α was downregulated after siTDP-43 treatment [101], indicating that the mechanism occurring during TDP-43-mediated TNF-α regulation is dependent on cell types, and likely involved the participation of other cellular factors. Indeed, these results highlight the importance of cell-specific differences in the shaping of TDP-43 functional properties [102].

On the other hand, ICAM-1 is an adhesion molecule that participates together with vascular cell adhesion molecule 1 (VCAM-1) at both interaction and extravasation of leukocytes at the BBB. ICAM-1 is also expressed in microglial cells and astrocytes of white and grey matter and its overexpression on CNS vascular endothelium is one of the hallmarks of brain inflammation [103]. In particular, it has been demonstrated that TDP-43 silencing is capable of upregulating *ICAM-1* mRNA expression in neuronal-like cells, as well as that the knockout of ICAM-1 in mice can confer resistance of encephalitis related to WNV infection, by diminishing viral load, leukocyte infiltration, and neuronal damage with respect to control animals [99,104]. Increasing levels of ICAM-1 were also reported in aging, bipolar disorder, and even dementia [105,106].

Notably, downregulation of TDP-43 was also associated with strongly suppression of IL-6 production after IL-1β and TGF-β1 stimulation in cultured primary human brain pericytes. Scotter and collaborators have hypothesized that this could be due to the TDP-43 dependent splicing regulation of hnRNP D [107]. These results have been supported by two recent publications regarding the role of Protein tyrosine phosphatase 1B (PTP1B) inhibition in attenuating astrocytes cell death as a consequence of TDP-43 mediated suppression of IL-6 [108], as well as the role of TDP-43 as a scaffold protein of the interleukin-6 and -10 splicing activating compartment (InSAC) [109].

Regarding the network TDP-43/virus/neurodegeneration, it is important to notice that IL-6 is an interleukin produced by several types of brain cells in way that is dependent on cell ages and in response to injury, such as tissue damage and infection [110]. Furthermore, IL-6 expression was found to be linked to both pro- and anti-inflammatory effects, highlighting its importance in the regulation of immune response. In keeping with these effects, IL-6 was also described to inhibit the replication of HBV via several mechanisms, including downregulation of human liver bile acid transporter Na(+)/taurocholate cotransporting polypeptide (NTCP) receptor [111], repression of viral transcripts by targeting the epigenetic modification (histone acetylation) of the HBV covalently closed circular DNA (cccDNA) [112] and by blocking the expression of both hepatocyte nuclear transcription factors (HNF) 1 and 4 alpha [113]. Circulating levels of IL-6 have also been used as predictive biomarker to identify severe SARS-CoV-2 pathology [114] and the increased levels of IL-6 were found to correlate with several abnormalities in blood tests observed in patients suffering from COVID-19 [115], as well as in HIV patients developing autoimmune disease [116].

Regarding neurodegeneration, increased levels of IL-6 expression were detected in the astrocyte-derived extracellular vesicles of sporadic ALS patients [117], in nigrostriatal region and in the cerebrospinal fluid of PD patients [118] and within and nearby amyloid plaques in AD patients [119]. On the contrary, the knockout of IL-6 in a mice transgenic model of Huntington’s disease (HD) was reported to worsen the phenotype associated to HD leading to dysregulation of genes essential for synaptic functions and relevant for the pathogenesis of HD [120].

Finally, neuroinflammatory pathways linked to TDP-43 dependent modulation of IL-1β and nuclear factor-κB (NF-κB) can also be observed during neurodegenerative disorders and viral illness. In 2020, Lee and collaborators demonstrated that overexpression of TDP-43 was able to induce IL-1β and NF-κB upregulation in primary mouse astrocytes. In this report, the signaling cascade TDP-43-PTP1B-NF-κB was found to be responsible for the TDP-43 mediated neuronal death and mitochondrial dysfunction observed in those cells [108].

In this respect, it is interesting to note that IL-1β is an immune mediator involved in pro-inflammatory signaling in astrocytes and glia cells [121,122], while NF-κB is a transcriptional factor of several cytokines, chemokines and is important in the functional activity of inflammasome [123].

Accordingly, increased levels of IL-1β were described in brain of AD patients [77] and it has been described to drive ALS pathogenesis in mice [124], as well as to promote SARS-CoV2 viral entry in A549 human lung cell line [125].

Furthermore, NF-κB resulted to be activated in ALS patients and mouse models [126,127] in a mouse model of progranulin (*GRN*)-deficient FTD [128], in PD patients [129], in HD patients and mouse models [130]. Moreover, a role of NF-κB signaling was detected in SARS-CoV-2 [131], HBV [132], HIV-1/2 [133,134], CVB3 [135], and HSV-1 [136] infection.

Last but not the least, it is important to mention the link between several long noncoding RNAs (lncRNAs), RBPs activity, and the role this can play in the maintenance of immune homeostasis [137,138]. In this context, metastasis associated lung adenocarcinoma transcript 1 (*MALAT1*) and nuclear paraspeckle assembly transcript 1 (*NEAT1*) are two lncRNAs implicated in neurodegenerative process and viral infection, also in association with TDP-43 [139,140,141,142,143]. For example, upregulation of *MALAT1* was reported in response of flavivirus and human papilloma virus (HPV) infection [141,142]. Additionally, a role of MALAT1 and TDP-43 in regulating the expression of antiviral type I IFN (IFN-I) has been recently proposed [143]. Specifically, in this work it has been demonstrated that different viruses were able to induce downregulation of *MALAT1* leading to TDP-43 release, activation, and consequently IFN-I production [143].

In addition, altered expression of *NEAT1* was described during HIV-1 and HSV-1 infection [144,145], as well as during several neurological disorders linked to TDP-43 disorders such as ALS and FTLD [140]. Moreover, upregulation of constitutive NEAT1 isoform (*NEAT1_1*) was reported to ameliorate TDP-43 induced toxicity in models of TDP-43 proteinopathy [139].

Taken together, all this information provides clear evidence of the importance of maintaining a healthy brain homeostasis, as well as supporting the role of TDP-43 and, more, in general, of RBPs in controlling transcripts implicated in neuronal inflammation and brain dysfunctions (Figure 1).

## 4. Conclusions

Over the years, the role of TDP-43 in the onset and progression of several disease have been extensively study. Despite the well-known involvement in brain pathologies, including, ALS, FTLD, Alzheimer’s, and Parkinson’s disease, TDP-43 has also been associated with cancer and viral infection [146]. Here, we summarized all the novelty regarding the interaction of TDP-43 and various DNA and RNA viruses (Figure 2). Overall, TDP-43 was found to actively contribute to viral entry, replication, and latency by binding several viral transcripts, as well as to participate in the establishment or at least significant contributions to neuropathic symptoms similar to those observed in neuronal disorders. Not by chance, some of the viruses presented in this review have also been described to be associated with brain dysfunction, as reported in Table 1. In keeping with this, during viral infections TDP-43 expression results to be frequently dysregulated, and it can also be post-translationally modified and translocated in the cytoplasm in a way that is similar to its modifications in neurodegeneration. All these alterations could therefore determine to the loss of TDP-43 endogenous functions, leading to cellular impairment and even cell death. It is interestingly to mention that TDP-43 collaborates with other RBPs in the maintenance of cell homeostasis and, presumably not by chance, some of these factors are also implicated in the same viral infection and in brain dysfunctions. For example, mutations in FUS encoding gene can exacerbate the sensibility to HIV-1 infection, and vice versa HIV-1 can increase the cytoplasmic localization of Fused in sarcoma (FUS) [147]. Furthermore, depletion of hnRNP K was described to reduce the HBV viral load in human hepatoma HepG2 cells, as well as hnRNP A1 was found to stimulate EV viral translation [27,60]. Finding new viral TDP-43 interactors could therefore be useful not only for a deeper characterization of the importance of TDP-43 pathology in viral disease, but also to provide deeper insight in the way this protein can establish unhealthy neurological conditions and inflammatory symptoms. As a consequence, the modulation of TDP-43 through chemical compounds, peptides, small interfering RNA (siRNAs), circular RNAs (circRNAs), and CRISPR/Cas9 based approaches may potentially represent novel therapeutic strategies for treating these virus-related disorders. Regarding TDP-43, in fact, several compounds have already been tested by employing different cellular/animal models and aspects of TDP-43 pathology, which includes its expression, nucleo-cytoplasmic balance and aggregation propensity [148]. For example, a small molecule called nTRD22, able to interact with the N-terminal domain of TDP-43 and to ameliorate the climbing defects, was observed in a *Drosophila* model of TDP-43 overexpression [149]. In the future, some of these compounds could be tested in cellular/animal models of viral infections to test their therapeutic functionality. In parallel, siRNA molecules and CRISPR/Cas9 therapeutic approaches aimed against specific TDP-43-viral protein interactors could also be tested to assess their ability to revert aberrant TDP-43 pathology during viral infections, as has been tried for neurodegenerative processes [150,151]. Finally, different antisense oligonucleotides (ASOs) have also been designed to target RBPs or their modifiers looking at neurodegenerative disorders and cancer. Among these, it is interesting to mention a study of 2017 performed on TDP-43 transgenic mice using ASOs targeting Ataxin-2 (Atxn-2) [152]. Basically, the reduction of Atxn-2 levels obtained by a single administration of ASOs into the CNS was successful in prolonging mice lifespan and improving their motor functions [152]. It would be therefore interesting to see, in the future, if some of these modulating factors of TDP-43 toxicity could also have a modifier effect on specific viral infections.

In conclusion, all the evidence that has been accumulating with regards to the increasing connection between TDP-43 and viral infections could represent a novel and potentially interesting area to better understand the role played by this protein in cellular metabolism, its possible connections with neurodegeneration, but also a novel area to explore to develop novel anti-viral therapeutic approaches.

## Figures and Tables

**Figure 1 ijms-24-01581-f001:**
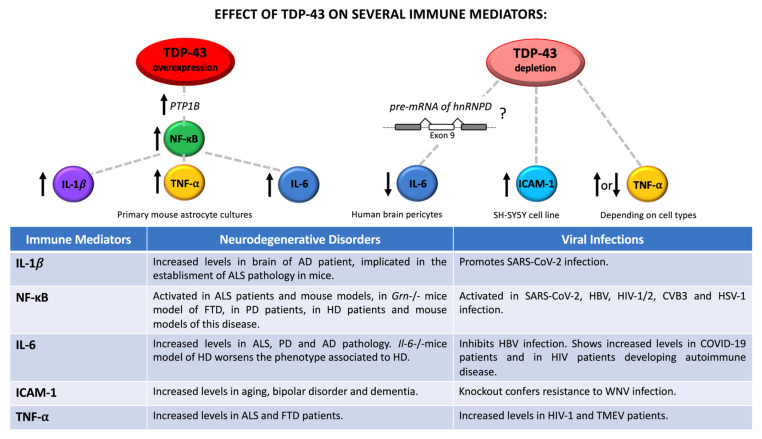
Relationship between TDP-43-controlled immune mediators and neuronal dysfunctions/viral infection. Regulation of TDP-43 levels contributes to increase or reduce the expression of different immune mediators, such as IL-1β, NF-κB, IL-6, ICAM-1, and TNF-α. Up- and downregulated effects are indicated with upward and downward arrows, respectively. Neurodegenerative and viral implications are also reported in the table.

**Figure 2 ijms-24-01581-f002:**
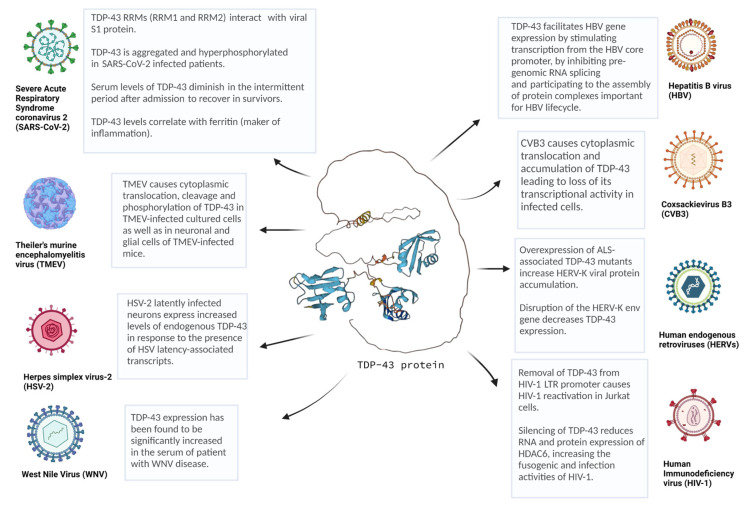
Overview of several viruses linked to TDP-43 proteinopathy. TDP-43 is involved in several infections by DNA and RNA viruses, including Coxsackievirus B3 (CVB3), Theiler’s murine encephalomyelitis virus (TMEV), human immunodeficiency virus (HIV), human endogenous retroviruses (HERVs), hepatitis B virus (HBV), severe acute respiratory syndrome coronavirus 2 (SARS-CoV-2), West Nile virus (WNV), and herpes simplex virus-2 (HSV). This picture was created with Biorender.com. Predicted TDP-43 ternary-structure was created using AlphaFold algorithm and reprinted with the permission from Refs. [153,154].

**Table 1 ijms-24-01581-t001:** List of TDP-43 related viruses and brain involvement.

Virus	Brain Involvement
Coxsackievirus B3 (CVB3)	CNS of neonatal mice were found to be susceptible to CVB illness likely through the infection of progenitor cells [155]. CVB3 was found to be associated with aseptic meningitis in a Hong Kong population [156].
Theiler’s murine encephalomyelitis virus (TMEV)	TMEV is widely used as a model to study multiple sclerosis [39]. It can induce apoptosis, neuronophagia and inflammation in infected neurons [157].
Human immunodeficiency virus (HIV)	HIV is the etiological agent of HIV encephalitis. HIV can be carried into CNS through infected CD4+ T cells and/or monocytes and, as a result, brain macrophages and microglia are widely considered the reservoirs for persistent viral infection. When these cells are activated, they can trigger an immunological response leading to neuronal death and the consequent establishment of HIV-associated dementia (HAD) [158]. HIV is responsible for opportunistic infection in CNS of HIV-positive individuals, including cerebral toxoplasmosis, progressive multifocal leukoencephalopathy (PML), tuberculous meningitis, cryptococcal meningitis and cytomegalovirus infection [159].
Human endogenous retroviruses (HERV)	HERV endogenous expression can be induced by viral RNAs and proteins, following infection of HIV-1, HBV and influenza A viruses [160,56]. Interestingly, increased HERV-K expression has been detected in brain of ALS patients, and it was found to induce neuronal injury in a model of transgenic animals [56].
Hepatitis B virus (HBV)	Impairment of neuronal structures has been observed to occur in HBV-positive individuals with associated chronic liver dysfunction [161].
Severe acute respiratory syndrome coronavirus 2 (SARS-CoV-2)	Cognitive dysfunctions and encephalitis were reported in patients suffering from SAR-CoV-2 infection [162,163]. SARS-CoV-2 has been proposed to induce morphological and cellular alteration of brain structures [162].
West Nile virus (WNV)	WNV causes severe neurological illness, generally referred to West Nile neuroinvasive disease (WNND) and including West Nile encephalitis (WNE), West Nile meningitis (WNM), and West Nile paralysis (WNP) [164]. WNV infection has been linked to neuronal dysfunction, loss of synapses, and astrocytic gliosis, both in human patients and animal models [165,166,167,168].
Herpes simplex virus-2 (HSV-2)	HSV-2 has been observed to cause neurological complication and establishes latent infection in neurons of ganglia [74,169].

## Data Availability

Not applicable.

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
