# Peer review of "Reviewing the Potential Links between Viral Infections and TDP-43 Proteinopathies"

_ijms, 2023, doi:10.3390/ijms24021581_

Round 1
Reviewer 1 Report
The manuscript from Dr. Rahic and colleagues offers a critical overview on the possible involvement of the RNA binding protein TDP-43 protective function against viral entry and infections in neurodegenerative diseases.
The manuscript is interesting and well written, and I do not have major concerns. I would only suggest to implement discussion on how these findings could be exploited for therapeutic purpose.
Moreover, I would suggest to mention the possible involvement of lncRNAs, in particular MALAT1, as regulators of TDP-43 antiviral interferon production and maintenance of immune homeostasis.
The title of the manuscript could be ameliorated conveying the main topics of the study.
Author Response
Point 1: I would only suggest to implement discussion on how these findings could be exploited for therapeutic purpose.
Response 1: We thank the Reviewer for this suggestion. We expanded our conclusions introducing some consideration (chapter 4 “Conclusions” lanes: 644-664) regarding different therapeutic strageties currently used to target RBPs in neurodegeneration and how they could also be employed to develop novel viral therapeutic approaches.
Point 2: Moreover, I would suggest to mention the possible involvement of lncRNAs, in particular MALAT1, as regulators of TDP-43 antiviral interferon production and maintenance of immune homeostasis.
Response 2: This is a very interesting suggestion and therefore in this revised version we have introduced a paragraph (chapter 3 ” The role of neuroinflammation in viral infection and neurodegeneration: a potential link with TDP-43 dysfunction” lanes: 547-562) related to long-non coding RNAs (specifically NEAT1 and MALAT1) and their involvement in TDP-43 pathology and viral infections.
Point 3: The title of the manuscript could be ameliorated conveying the main topics of the study.
Response 3: As required we modified our title as follows: “Reviewing the potential links between viral infections and TDP-43 proteinopathies”.
Reviewer 2 Report
This review provides a comprehensive description of relationship between viral infections and neurodegenerative diseases including ALS, focusing on the RNA-binding protein, TDP-43.
The review starts with a topic about ALS-like symptoms due to Enteroviruses infection, and RBP hijacking by the virus transcripts. And it summarizes various knowledge about associations between infections of several virus and neurodegenerative diseases and RBP regulations.
Mechanisms of the associations were mainly classified into two processes: 1) disruption of RBP balance due to binding of RBPs to viral transcripts and 2) changes in RBP balance in response to host immune responses. Among them, the first process is a phenomenon in cells in the brain such as neurons and astrocytes. One of the few disadvantages of this review is that it is hard to find which viruses have been reported to infect into the brain cells. I think a figure or table summarizing this point must help a lot for systematic understanding of this review.
Author Response
Point 1: Mechanisms of the associations were mainly classified into two processes: 1) disruption of RBP balance due to binding of RBPs to viral transcripts and 2) changes in RBP balance in response to host immune responses. Among them, the first process is a phenomenon in cells in the brain such as neurons and astrocytes. One of the few disadvantages of this review is that it is hard to find which viruses have been reported to infect into the brain cells. I think a figure or table summarizing this point must help a lot for systematic understanding of this review.
Response 1: We thank the Reviewer for this suggestion. We have now prepared a new table (referred to Table 1, chapter 4 “Conclusions”) highlighting the brain involvement of the TDP-43 related viruses considered in this review.